# Is Intermittent Abdominal Pressure Ventilation Still Relevant? A Multicenter Retrospective Pilot Study

**DOI:** 10.3390/jcm12072453

**Published:** 2023-03-23

**Authors:** Valeria Volpi, Eleonora Volpato, Elena Compalati, Paola Pierucci, Antonello Nicolini, Agata Lax, Laura Fagetti, Anna Annunziata, Rosa Cauteruccio, Giuseppe Fiorentino, Paolo Banfi

**Affiliations:** 1IRCCS Fondazione Don Carlo Gnocchi ONLUS, 20148 Milan, Italy; 2Department of Psychology, Università Cattolica del Sacro Cuore, 20123 Milan, Italy; 3Cardiothoracic Department, Respiratory and Critical Care Unit, Bari Policlinic University Hospital, 70121 Bari, Italy; 4Section of Respiratory Diseases, Department of Basic Medical Science Neuroscience and Sense Organs, University of Bari ‘Aldo Moro’, 70122 Bari, Italy; 5Department of Respiratory Pathophysiology and Rehabilitation Monaldi-A.O. Dei Colli, 80131 Naples, Italy; 6Department of Intensive Care, Azienda Ospedaliera di Rilievo Nazionale dei Colli, 80131 Naples, Italy

**Keywords:** intermittent abdominal pressure ventilation (IAPV), non-invasive ventilation (NIV), neuromuscular diseases (NMD), retrospective study, amyotrophic lateral sclerosis (ALS), Pompe disease, Duchenne muscular disease (DMD), quality of life

## Abstract

Non-invasive ventilatory support (NVS) is a technique used to reduce respiratory work in neuromuscular diseases, preventing the progression of respiratory failure. NVS is usually administered via a nasal or an oronasal mask, causing discomfort, especially in patients ventilated for more than 16 h/day. Intermittent abdominal pressure ventilation (IAPV) differs completely from conventional NVS and consists of a portable ventilator and a corset with Velcro closures as the interface. In our study, the practicability and efficacy of IAPV were studied in three Italian centers monitoring 28 neuromuscular patients using IAPV who were then retrospectively analyzed. The primary outcomes were an improvement in hypoxemia and the normalization of hypercapnia, and the secondary outcome was an improvement in quality of life. Data were collected at baseline (T0) and after two hours of ventilation (T1), with follow-ups at three months (T2) and six months (T3). Statistical significance was found for PaCO_2_ over time (F (2.42) = 7.63, *p* = 0.001) and PaO_2_ (W = 0.539, *p* = 0.033). The time of NVS usage also significantly affected the quality of life (F (2.14) = 6.90, *p* = 0.010), as seen when comparing T0 and T3. As an alternative ventilation method, IAPV is still relevant today and could become a key part of daytime support, especially for patients who do not tolerate standard daytime NVS with an oral interface.

## 1. Introduction

Non-invasive ventilator support (NVS) is a well-established treatment that is used to reduce respiratory work due to muscle weakness in neuromuscular diseases (NMD), preventing the progression of respiratory failure to intubation and/or tracheostomy [1]. Initially being used mainly during the night, as the disease progresses NVS is also frequently used during the day [2,3]. It is usually delivered via a nasal or an oronasal mask, which can cause discomfort and/or cosmetic problems, leading to poor patient compliance/adherence [1].

Intermittent abdominal pressure ventilation (IAPV) is a valid, unconventional alternative to daytime NVS [4]. It consists of a portable ventilator and a corset equipped with an internal bladder as an interface. The corset is lightweight, comfortable and, due to its Velcro fasteners, easy to wear. Cyclic inflation of the bladder moves the diaphragm upwards to expel air up to the residual volume. When the bladder is deflated, the abdominal viscera and the diaphragm are lowered by gravity, and inhalation then occurs passively due to the elastic return of both the lungs and rib cage and the descent of the diaphragm itself. The bladder is connected via a flexible circuit to a portable ventilator with an internal battery that is capable of delivering up to 2.5 L of air into the abdominal wall.

IAPV is indicated for neuromuscular disease and has already been tested in some preliminary studies and case reports. A narrative review by Pierucci et al. underlined the advantages of IAPV as a useful alternative for daytime support [5]. IAPV contributes to improved gas exchange, reduced symptoms and improved quality of life by reducing the incidence of pneumonia and avoiding the need for intubation and tracheotomy [6,7]. 

Due to the paucity of data and the small sample sizes of the existing studies [6,8,9], we conducted a retrospective pilot study to evaluate the practicability and efficacy of IAPV application in our centers. These data will be useful for efforts to design a randomized controlled trial that will evaluate the effectiveness of IAPV intervention in neuromuscular diseases (NMD). 

## 2. Materials and Methods

### 2.1. Study Design and Participants

A multicenter retrospective pilot study was conducted at the following centers: IRCCS Santa Maria Nascente, Fondazione Don Carlo Gnocchi, in Milan, Italy; Department of Respiratory Pathophysiology and Rehabilitation Monaldi-A.O. Dei Colli in Naples; Cardiothoracic Department, Respiratory and Critical Care Unit, Bari Policlinic University Hospital; and the Section of Respiratory Diseases, Department of Basic Medical Science, Neuroscience and Sense Organs, “Aldo Moro” University of Bari. 

Patients treated with IAPV between 2019 and November 2022 were identified retrospectively from the patients’ records. 

Participants were included if they met one or more of the following criteria: (I) a diagnosis of neuromuscular disease (NMD), (II) ongoing non-invasive interface intolerance, (III) clinical stability and (IV) age ≥18 years, having been on NVS >16 h/day for at least one year. The exclusion criteria were as follows: acute respiratory failure in the past month, the presence of invasive ventilation, severe bulbar weakness (ALSFRS-R Bulbar score < 9), the presence of kyphoscoliosis, abdominal or thoracic intervention, and refusal to participate in the study.

### 2.2. Interventions

In our study, IAPV interventions were delivered using the LunaBelt (Dima Italia Inc., Bologna, Italy), a portable ventilator with an internal battery, and the PBelt corset as an interface adapted to the abdominal shape of each subject. Before use, the following parameters were set and adjusted for each patient: Inspiratory Positive Airway Pressure (IPAP), inspiratory time, rise time and Respiratory Rate (RR). 

A functional respiratory assessment was performed at baseline with spontaneous breathing and two hours after IAPV. Patients were discharged with instructions to continue daytime ventilation at home with IAPV for a prescribed number of hours. 

### 2.3. Outcome Measures

The primary outcome measures were changes in the respiratory parameters in terms of an improvement in hypoxemia of at least 5 mmHg and the normalization of hypercapnia (arterial pressure of carbon dioxide in the normal range), as assessed by blood gas analysis (BGA) during IAPV.

The secondary outcome was an improvement in quality of life (QoL).

### 2.4. Measurements

Data were extracted from the electronic medical records using a data extraction form. The data retrieved consisted of the following patient demographics: sex, age and primary diagnosis. Four assessment times were considered: the baseline (T0), after two hours of ventilation (T1) and follow-ups after three months (T2) and six months (T3). The time before the start of treatment with IAPV was set as the baseline (T0). 

The data extracted at baseline were as follows: -Spirometry: Forced Vital Capacity (FVC), Forced Expiratory Volume in the first second (FEV_1_) and Tiffeneau Index (FEV_1_/FVC), obtained through the Master Screen Body Jaeger Vyntus™ Pneumo, Vyaire, Mettawa, IL, USA.-IAPV settings for the expiratory and inspiratory volumes and respiratory frequency, obtained via Citrex H4, DIMA Italia, Bologna [10].-Strength of respiratory muscles: Maximal Inspiratory Pressure (MIP) and Maximal Expiratory Pressure (MEP), obtained using a MicroRPM Pressure Meter, Micro Medical Ltd., Lewiston, ME, USA.-Blood gas analysis (BGA) (pH (measured acid-base balance of the blood); partial pressure of oxygen (PaO_2_); Partial pressure of carbon dioxide (PaCO_2_); and bicarbonate (HCO_3_^−^)), obtained using the GEM 3500, Instrumentation Laboratory, Milan, Italy.-Quality of life (QoL) was assessed by the administration of the Short Form-12 Questionnaire (SF-12) [11], a shortened version of its predecessor, the SF-36 [12], which, in turn, evolved from the Medical Outcomes Study. After two hours (T1) of ventilation with IAPV, the following variables were investigated:-Respiratory Rate (RR).-Inspiratory/expiratory volume, measured for one minute during spontaneous breathing and 10’ after starting IAPV for another minute.-BGA.

BGA results were also collected during the follow-up after three months (T2) and after six months (T3) from electronic record data. The parameters collected after three months (T2) were as follows:-Spirometry (FVC, FEV_1_, FEV_1_/FVC).-Inspiratory/expiratory volume, measured for one minute during spontaneous breathing and 10’ after starting IAPV for another minute.-Quality of life.

At six months from baseline (T3), the following data were assessed:
-Spirometry.-Quality of life.

### 2.5. Statistical Analysis

Statistical analysis was performed using the statistical software Jamovi (version 2.3.21). Continuous data were expressed as means and SDs and categorical data were expressed as percentages and absolutes. If data were normally distributed, repeated-measure analysis of variance (ANOVA) was performed to follow changes in values during the study. Friedman’s test was considered for non-normally distributed data. The post hoc testing utilized Tuckey’s test.

For normally distributed data, the 95% confidence interval (CI) is stated. Statistical significance was assumed for a *p*-value of ≤0.05.

## 3. Results

The overall sample consisted of 28 people with neuromuscular disease (mean age = 54.5 (±16.3); 35.7% female), recruited from the above-mentioned centers in the period between 2019 and November 2022 (Figure 1). Their pathologies were distributed as follows: amyotrophic lateral sclerosis (19–67.8%), Duchenne muscular dystrophy (5–17.8%), Pompe disease (2–7.1%), limb girdle muscular dystrophy (1–3.5%) and mitochondrial myopathy (1–3.5%) (Figure 2). 

Table 1 shows the main socio-demographic and clinical characteristics of the sample considered.

The setting variations and measurements of gas exchange made after the initiation of IAPV are shown in Table 2 and Table 3. The IAPV parameters (timed modality) were the PBelt pressure 51 ± 10 hPa, inspiratory time 1.5 ± 0.12 s, Respiratory Rate 14 ± 1.35 bpm and rise time 0.4 ± 0.2 s.

Since we had a small sample size (*N* = 28), determining the distribution of the variables PaO_2_, PaCO_2_ and the time of IAPV usage were important in order to choose an appropriate statistical method. Therefore, Mauchly’s test of sphericity was performed for each primary outcome. Mauchly’s test of sphericity indicated that the assumption of sphericity was not violated for PaCO_2_ (W = 0.659, *p* = 0.078). Conversely, Mauchly’s test was statistically significant for PaO_2_ (W = 0.539, *p* = 0.033) and the time of IAPV usage (W = 0.441, *p* = 0.007). For these two variables, the Greenhouse–Geisser value was <0.75, and we then considered the Greenhouse–Geisser correction. Based on these results, and after a visual examination of the histogram and the QQ plot, we decided to proceed in conducting a parametric test. Moreover, the mean with the standard deviation was used to summarize the variable.

A repeated-measure ANOVA was performed to compare the effect of the time of IAPV usage on PaCO_2_. There was a statistically significant difference in PaCO_2_ over time (F(2.42) = 7.63, *p* = 0.001). Tukey’s post hoc difference tests indicated that there was a significant improvement in PaCO_2_ over time, even if there was no significant difference during the last three months between T2 and T3. The range of monitored PaCO_2_ during the first 2 h study period for all the patients was from 50.1 (±5.33) to 39.6 (±3.25) mm Hg.

Another repeated-measure ANOVA was performed to compare the effect of the time of IAPV usage on PaO_2_. There was not a statistically significant difference in PaO_2_ over time (F (2.12) = 2.94 *p* = 0.060, Table 4 and Figure 3).

Finally, if we compare the baseline (T0) and T3, the time of IAPV usage significantly affected the QoL outcome, as assessed using the SF-12 questionnaire (F2.14 = 6.90, *p* = 0.010). However, no notable differences were found over the last three months, that is, comparing T2 and T3 for both the Physical Component Summary and the Mental Component Summary (Figure 4).

## 4. Discussion

Non-invasive ventilator support (NVS) is a technique of ventilation that normally uses a mask (nasal, oronasal or facial) or a helmet, in the acute setting, to deliver positive pressure into the airway. One of the earliest descriptions of the use of NVS with a nasal mask dates back to 1987, when it was used to treat nocturnal hypoventilation in patients with neuromuscular diseases [13,14]. In recent years, its popularity has grown enormously, and there has been a proliferation of trials and meta-analyses on its use [15,16]. NVS offers numerous benefits by reducing the need for orotracheal intubation and its related side effects such as the trauma of the upper airway, impairment of speech and swallowing and, especially, pneumonia associated with a ventilator. Pneumonia has an incidence of about 9–27% among all mechanically ventilated patients, with a mortality rate recently estimated at 9–13% in various studies [17,18,19,20]. In neuromuscular and/or kyphoscoliotic patients, NVS improves gas exchange and reduces the rate of intubation in cases of acute respiratory failure [21,22]. In our opinion, however, in neuromuscular patients on ventilation for more than 16 h per day, different types of interfaces need to be offered to avoid adverse events as much as possible, which is why we adopted the IAPV [23,24]. In our study of 28 patients, 13 had nasal decubitus; 1 patient had a nuchal decubitus; 2 began to have claustrophobia problems at the oronasal interface; 3 were engaged in smart working and did not want to be seen on video with the nasal interface; 2 preferred IAPV to mouth-piece ventilation; and 7 used IAPV during ambulation in preference to the nasal interface.

The intermittent abdominal pressure ventilation mode differs completely from the conventional non-invasive ventilation modalities that provide active exhalation rather than inspiration. 

Since its invention and first application for a limited availability of steel lungs during a poliomyelitis outbreak, IAPV has undergone several adjustments leading up to the achievement of its present-day composition, represented by an abdominal corset or Pneumobelt^®^. In the last few years, this modality of ventilation has reappeared on the European market [3,25,26,27,28,29]. 

In our study, we used a new model of the abdominal corset that is less bulky, lighter, easy to wear and tailor-made for the patient and uses Velcro for abdominal fixation (Pbelt). 

To inflate the corset, we used a home ventilator specifically designed to operate with IAPV through dedicated software (LunaBelt, Dima Italia, Bologna). The LunaBelt is a turbine ventilator with an internal battery and can also be used for non-invasive respiratory support with different interfaces. The effectiveness of IAPV depends mainly on the area of the chest and abdomen that is covered by the band and the position the patient has adopted; therefore, the patient must be seated [3] or at least in a semi-recumbent position with an angle of 30° or greater [1]. The use of the abdominal corset is comfortable and simple to use, and above all, it allows the face, mouth, nose and neck to be left free. It is portable and can therefore be easily installed on a wheelchair. As confirmed by our study, IAPV can improve the quality of life of individuals with neuromuscular disease by increasing their ability to breathe and consequently allowing them to participate in more daily activities. While using the IAPV, it is possible to speak, walk and cough more effectively on account of an increased expiratory flow due to abdominal compression, promoting the removal of secretions. It is also useful for constipation [3,5]. In addition, it clears the face and allows the patient to have a more normal sense of smell, eliminating facial or nasal interfaces that can be colonized by pathogenic bacteria. Oral nutrition is also possible [3], and previous papers showed that enteral nutrition can also be managed [3]. Yang et al. described its use overnight in a semi-sitting position [18]. 

Disadvantages include the regurgitation of food during meals (though this occurs rarely), the entanglement of clothing with straps or Velcro closures and the inability to shower or bathe during its utilization. Regular follow-ups are important because usage over time may become less effective and should therefore be evaluated [18,19]. 

In our study, both blood gases and ventilation parameters were monitored, and we saw that IAPV improved the data, relieved dyspnea and continued to be a comfortable daytime alternative to NVS for patients ventilated for more than 16 h a day. Another model of NVS features a mouthpiece that allows varying tidal volumes so that active lung volume recruitment (air stacking) can be performed [30]. However, a major drawback remains for those patients with neuromuscular disorders who cannot properly grasp a mouthpiece (MPV), and nasal NVS may be ineffective because air escapes from the mouth, resulting in useless ventilation. Nardi et al. demonstrated that MPV can cause states of hypoventilation when the patient eats or answers the phone, for example, but the results of this hypoventilation have not yet been evaluated [31]. 

For all the reasons stated above, IAPV is becoming an important alternative for neuromuscular patients on ventilation, even during daylight hours. Therefore, for some neuromuscular patients who require both daytime and night-time NVS, the only remaining options to avoid orotracheal intubation and tracheotomy are the reliance on masks at night and IAPV during the daytime. 

Among the limitations of the study is the low sample size, which limits the possibility of generalizing the results. Another relevant limitation is the lack of a comparison between NVS and IAPV according to a randomized and controlled approach, which would have also helped to highlight a comparison between the criticalities of one device and the other. 

Additionally, the use of IAPV can be costly and may require frequent replacement of equipment or supplies, thus making it an economic burden. It also requires a high level of care and maintenance, and the use of the device must be supervised by healthcare professionals, which can limit the individual’s autonomy and independence.

Finally, IAPV can be an effective way of improving the quality of life of individuals with neuromuscular diseases by helping them to breathe more easily, but it can also result in some discomfort and may be costly. However, IAPV remains the preferred option, as we have already mentioned, for aesthetic reasons.

## 5. Conclusions

Daytime NVS is essential for neuromuscular patients with advanced ventilatory failure and is effective in avoiding acute respiratory failure and early recourse to tracheotomy. We also know that patients autonomously extend night-time use of NVS to daytime hours. Different interfaces and the use of mechanical in-exhalation play key roles in the success of NVS. It is our opinion that with time, “alternative” ventilation methods could become a key part of daytime support. Further research in this emerging field is strongly warranted to further improve the quality of life and breathing of neuromuscular patients.

## Figures and Tables

**Figure 1 jcm-12-02453-f001:**
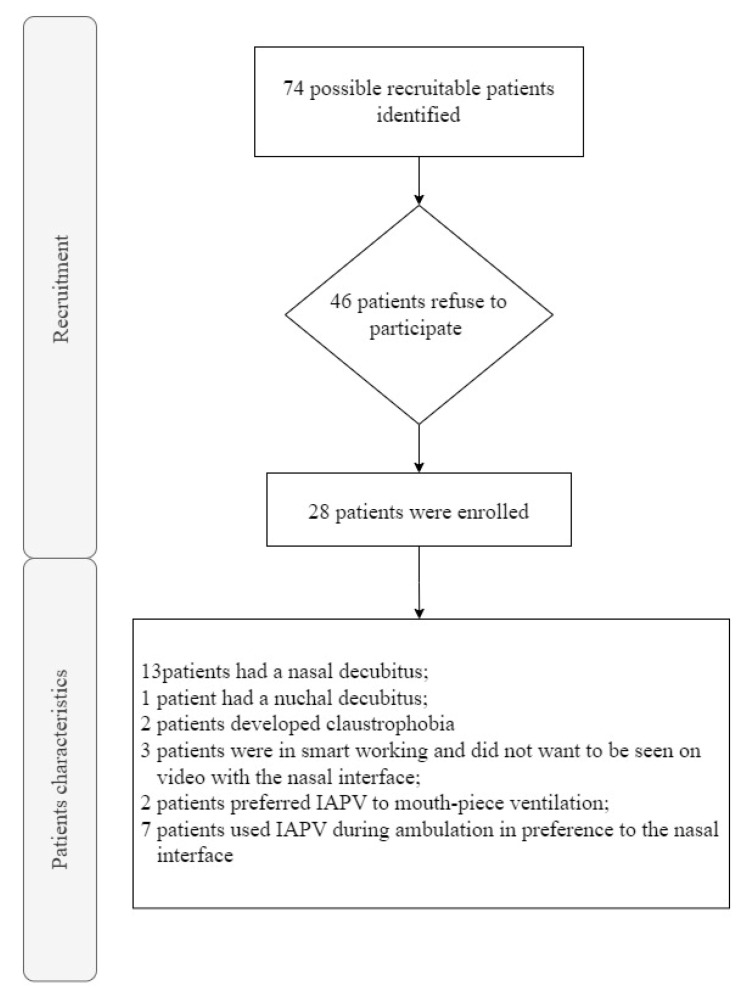
Flow chart of the participants’ selection.

**Figure 2 jcm-12-02453-f002:**
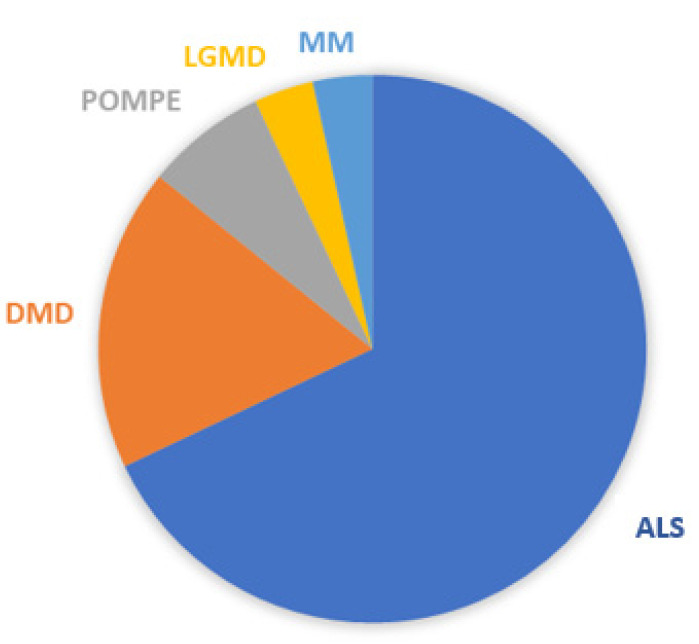
Distribution of pathologies.

**Figure 3 jcm-12-02453-f003:**
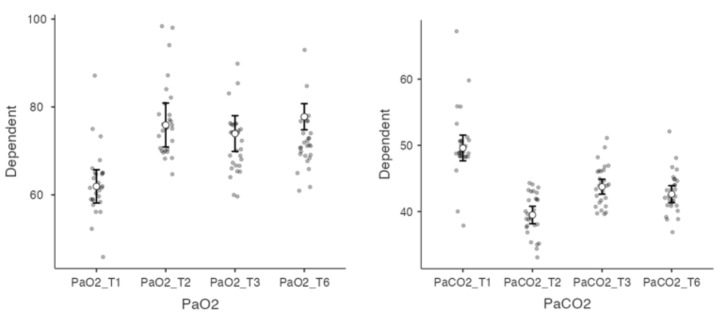
Pairwise Comparisons (Durbin–Conover). Notes: PaO_2_ = Partial pressure of oxygen; PaCO_2_ = Partial pressure of carbon dioxide; HCO_3_^−^ = Bicarbonate.

**Figure 4 jcm-12-02453-f004:**
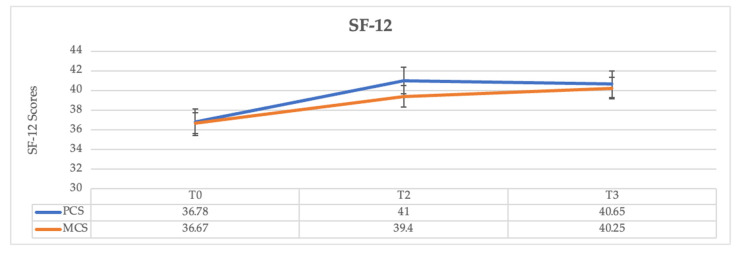
Development of scores from the SF-12 subscales before and after adaptation to IAPV. Notes: T0 = Baseline; T2 = 3 months; T3 = 6 months; PCS = Physical Component Summary; MCS = Mental Component Summary.

**Table 1 jcm-12-02453-t001:** Characteristics of patients at baseline.

Characteristics	Sample (*N* = 28)
Age	54.5 (±16.3)
Female, %	35.7 (±0.50)
Age at onset of NVS, yr	40.35 (±16.63)
NVS use, mo	139.5 (±255.8)
NVS use time (h/day)	18.3 (±2.54)
pH	7.41 (±0.03)
PaCO_2_ (mmHg)	50.1 (±5.33)
PaO_2_ (mmHg)	62.6 (±7.46)
HCO_3_^−^ (mmHg)	32.3 (±4.76)
FVC %	31 (±15)
FEV_1_%	33.8 (±15.1)
FEV_1_/FVC	89.7 (±9.06)
MIP	28.7 (±10.3)
MEP	32.1 (±10.8)

Results are shown as mean ± SD. FVC = Forced Vital Capacity; FEV_1_ = Forced Expiratory Volume in the first second; FEV_1_/FVC = Tiffeneau Index; MIP = Maximal Inspiratory Pressure; MEP = Maximal Expiratory Pressure; PaO_2_ = Partial pressure of oxygen; PaCO_2_ = Partial pressure of carbon dioxide; HCO_3_^−^ = Bicarbonate.

**Table 2 jcm-12-02453-t002:** Setting variations between baseline with spontaneous breathing and after two hours of ventilation with IAPV. Results are shown as mean ± SD. T0 = Baseline; T1 = After 2 h of IAPV.

	Spontaneous Breathing (T0)		IAPV after 2 h (T1)	
	Min	Max	Min	Max
Frequency	27 (±4.36)	32 (±4.22)	15.0 (±1.12)	15.0 (±1.12)
Inspiratory Volume	197 (±38.5)	254 (±49.4)	731 (±72.6)	802 (±75.1)
Expiratory Volume	196 (±38.9)	258 (±50.4)	730 (±79.4)	835 (±75.9)

**Table 3 jcm-12-02453-t003:** Blood gas analysis variations between baseline (T0) and follow-ups (2 h (T1), 3 months (T2) and 6 months (T3)).

	T0	T1	T2	T3
pH	7.41 (±0.03)	7.47 (±0.03)	7.40 (±0.02)	7.40 (±0.02)
PaCO_2_ (mmHg)	50.1 (±5.33)	39.6 (±3.25)	44.0 (±2.99)	42.85 (±3.12)
PaO_2_ (mmHg)	62.6 (±7.46)	76.9 (±8.63)	71.8 (±7.01)	72.32 (±7.01)
HCO_3_^−^ (mmHg)	32.3 (±4.76)	30.3 (±4.29)	28 (±2.90)	28.53 (±2.33)
MIP	28.7 (±10.3)	n.a.	32.2(±30)	n.a.
MEP	32.1 (±10.8)	n.a.	36.5 (35.5)	n.a.

Notes: Results are shown as mean ± SD. T0 = Baseline; T1 = After 2 h of IAPV; T2 = 3 months; T3 = 6 months; pH = measured acid-base balance of the blood; PaO_2_ = Partial pressure of oxygen; PaCO_2_ = Partial pressure of carbon dioxide; HCO_3_^−^ = Bicarbonate; RR = Respiratory Rate; MIP = Maximal Inspiratory Pressure; MEP = Maximal Expiratory Pressure; n.a. not available for all patients.

**Table 4 jcm-12-02453-t004:** Variation in blood gas analysis (BGA) results between different assessments.

	Variation	*p*
PaO_2__T0–PaO_2__T1	12.23	<0.001
PaO_2__T0–PaO_2__T2	7.79	<0.001
PaO_2__T0–PaO_2__T3	9.51	<0.001
PaCO_2__T0–PaCO_2__T1	13.77	<0.001
PaCO_2__T0–PaCO_2__T2	6.97	<0.001
PaCO_2__T0–PaCO_2__T3	8.42	<0.001
PaO_2__T1–PaO_2__T2	4.44	<0.001
PaO_2__T1–PaO_2__T3	2.72	0.007
PaCO_2__T1–PaCO_2__T2	6.79	<0.001
PaCO_2__T1–PaCO_2__T3	5.34	<0.001
PaO_2__T2–PaO_2__T3	1.72	0.087

Notes: T0 = Baseline; T1 = After 2 h of IAPV; T2 = 3 months; T3 = 6 months; PaO_2_ = Partial pressure of oxygen; PaCO_2_ = Partial pressure of carbon dioxide.

## Data Availability

Availability on request.

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
