# Peer review of "Is Intermittent Abdominal Pressure Ventilation Still Relevant? A Multicenter Retrospective Pilot Study"

_jcm, 2023, doi:10.3390/jcm12072453_

Round 1
Reviewer 1 Report
Review:
The article ‘’Is Intermittent Abdominal Pressure Ventilation still current? A multicenter Retrospective Pilot Study’’ is one of the articles which reminds us that neuromuscular disorders (NMD) are very complex diseases regarding the therapeutic approaches. Neuromuscular disorders are one of the indications where the non-invasive ventilation (NIV) is a cornerstone of the therapy. The Intermittent Abdominal Pressure Ventilation-IAPV (known from almost 100 years ago) is a type of non-invasive respiratory support which contributes to better gas exchange, better quality of life and therefore better outcomes in the patients with NMD. IAPV is also one of those, for some times abandoned, so called ‘’old fashion’’ therapeutic procedures, which appeared more than once as a solution for long non-invasive support and for better comfort in patients treated with NIV for most of the day. This article is a contribution to prove efficacy of IAPV with NMD, especially in patients with Amyotrophic Lateral Sclerosis (ALS) considering that most patients included in this study where from this group of NMD.
Comments:
1. The article is written in good manner, mostly understandable
2. The article provides precisely the informations about the devices used in this study
3. The part of the paper which explains which parameters were investigated in different assesement times is confusing, it should be presented in table
4. It cannot be concluded from the article were the patients in stable condition during the study e.g. from the moment when the IAPV was introduced
5. What kind of complications regarding the use of interface with NIV were present before the IAPV was introduced (in the article Fiorentino G, Annunziata A, Coppola A, et al. Intermittent Abdominal Pressure Ventilation: An Alternative for Respiratory Support.Can Respir J. 2021;2021:5554765. Published 2021 Aug 23. doi:10.1155/2021/5554765 in the part Materials and Methods the authors precisely described the reasons why the NIV via the mask or nasal canulla were refused or connected with the complications)
6. The paper doesn’t provide the information about the pressure set in the bladder and didn’t apostrophised what were the IAPV settings generally
7. Also, the authors didn’t mentioned whether the tracheostomy was the exclusion criteria
Suggestions: Regarding the QOL questionnaire used in this study, I think that disease specific Health-Related QOL (HRQOL) questionnaire would be more appropriate, for example respiratory-specific QOL questionnaire. Also, the degree of the dyspnea would be more appropriate clinical parameter for evaluation
Author Response
Dear Reviewer,
We updated the manuscript following your indications.
We remain at your disposal for any further requests or corrections,
Best Regards

Reviewer 2 Report
In this paper the Authors describe a cohort of neuromuscular patients treated with Intermittent Abdominal Pressure Ventilation. Overall, the paper addresses an interesting and relevant topic within the context of the management of respiratory failure of patients with neuromuscular diseases. However, in order to improve this manuscript there are some point that need to be better cleared and addresses
- - Within the context of this scientific paper, the Authors should avoid terms / adjectives that were not supported by the results of data. For instance, while is it correct to describe the technique and equipment for IAPV, the Authors should avoid to use such adjectives as “valid” or “comfortable” (Introduction), which are not supported by the data. Alternatively, the Authors should support each of these claims with appropriate reference to previous studies (if any)
- - Methods: the Authors should add a paragraph describing the criteria for patient’s selections. Indeed, it is not clear why those patients were treated with IAPV rather that other NIMV techniques. I think that this information would be very important in order to understand the features of the patients that were treated and could benefit of this type of approach.
- - Results: the Authors should report how many patients were screened and how many did not tolerate this approach. Since this is a retrospective study and this information is not provided in the text, the Authors should avoid to report in the abstract that “feasibility “ was studied.
- - Methods – Measures: most importantly, the Authors should report in the Results, adding a Table with complete descriptive statistics, all the data that were registered, consistently with the Methods described in paraghaph 2.4. An option would be to report all the data in Table 3.
- - Figure 1: “ASL” should be “ALS”
- - Table 3: see above: all data (as per Methods) should be reported here; moreover, a column should be added for statistical significance. Information about the number of patients for each measure (time/point ) should be added.
- Results – pag 5: the methodological considerations for statistical analysis (lines 173-182)are redundant and should be reported in the “Methods”
- Table 4: please add + of – for each difference
- Results – Quality of life: the results should be more clearly presented (see above): scales and subscales should be reported. I suggest adding a graph for QOL too.
- Discussion: please add a paragraph in order to clearly point out study limitations.
- Discussion: overall, the discussion appears in some parts over- emphatic, and not reflecting the data reported in this study. I suggest to point our which kind of patients could, according to the opinion of the authors (since there are no conclusive and not enough data on this topic), benefit from this approach, clearly pointing out that further studies are needed in order to more appropriately address this topic.
Author Response
Dear Reviewer,
we modified the manuscript following your indications.
We remain at your disposal for any further questions or requests
Best regards

Round 2
Reviewer 2 Report
The Authors have replyed to all the points previously raised